# Effect of Water Deficit on Morphoagronomic and Physiological Traits of Common Bean Genotypes with Contrasting Drought Tolerance

**Leonardo Godoy Androcioli [1,2], Douglas Mariani Zeffa [1,3,*] , Daniel Soares Alves [1], Juarez Pires Tomaz [1] and Vânia Moda-Cirino [1]**

[1]  Department of Plant Breeding, Instituto Agronômico do Paraná, Londrina 86047-902, Brazil; leonardoandrocioli@hotmail.com (L.G.A.); danielsoares31@yahoo.com.br (D.S.A.); juarez.pires.tomaz@gmail.com (J.P.T.); vmvamoci@gmail.com (V.M.-C.)
[2]  Department of Agronomy, Universidade Estadual Paulista "Júlio de Mesquita Filho", Botucatu 18610-034, Brazil
[3]  Department of Agronomy, Universidade Estadual de Maringá, Maringá 87020-900, Brazil
*  Correspondence: douglas.mz@hotmail.com; Tel.: +55-43-996594575

**Abstract:** Water deficit is considered one of the most limiting factors of the common bean. Understanding the adaptation mechanisms of the crop to this stress is fundamental for the development of drought-tolerant cultivars. In this sense, the objective of this study was to analyze the influence of water deficit on physiological and morphoagronomic traits of common bean genotypes with contrasting drought tolerance, aiming to identify mechanisms associated with tolerance to water deficit. The experiment was carried out in a greenhouse, arranged in a randomized complete block $4 \times 2$ factorial design, consisting of four common bean genotypes under two water regimes (with and without water stress), with six replications. The morphoagronomic and physiological traits of four cultivars, two drought-tolerant (IAPAR 81 and BAT 477) and two drought-sensitive (IAC Tybatã and BRS Pontal), were measured for 0, 4, 8, and 12 days, under water deficit, initiated in the phenological stage R5. Water-deficit induced physiological changes in the plants, altering the evaluated morphoagronomic traits. The drought tolerance of cultivar BAT 477 is not only a direct result of the low influence of water deficit on its yield components, but also a consequence of the participation of multiple adaptive physiological mechanisms, such as higher intrinsic water use efficiency, net photosynthesis rate, transpiration, carboxylation efficiency, stomatal conductance, and intracellular concentration of $CO_2$ under water deficit conditions. On the other hand, cultivar IAPAR 81 can be considered drought-tolerant for short water-deficit periods only, since after the eighth day of water deficit, the physiological activities decline drastically.

**Keywords:** *Phaseolus vulgaris* L.; abiotic stress; plant breeding

## 1. Introduction

Common bean (*Phaseolus vulgaris* L.) is considered one of the most important legume crops in the world [1]. It is a staple food for over 300 million people in Latin America and Africa and an important source of protein, fiber, carbohydrates, and trace minerals [2,3]. Common bean is grown predominantly by small farmers, often in marginal regions, where crop yields are frequently affected by a number of abiotic factors, in particular water deficit [4,5]. Worldwide, water scarcity is estimated to affect about 60% of the common bean production areas with prolonged drought periods [1]. Drought is believed to be one of the most influential factors for reduced common bean yields, destabilizing the security of production systems in developing countries [6].

The effects of water deficit on common bean have been extensively studied [7–9]. Drought severity is defined by the stress frequency, duration, and intensity and by the crop development phase it occurs in [10,11]. Water deficit reduces leaf stomatal conductance and root hydraulics, decreasing water and nutrient uptake and photosynthetic plant activity. Moreover, water stress can lead to flower abortion, pod drop, reduce the efficiency of biological nitrogen fixation, and consequently decrease grain yield [6,12]. Drought stress can cause yield losses of 10–90%, and in the worst-case scenario, total production losses [13,14].

Drought tolerance is a quantitative trait that is regulated by multiple genes and strongly influenced by environmental conditions [15–17]. The yield stability of water-stressed genotypes with a deeper root system and higher accumulated biomass production and translocation is usually greater [14,18,19]. The physiological traits stomatal density and conductance, transpiration and photosynthesis rates, and relative leaf water content are strongly associated with drought tolerance, as water-use efficiency, that is also a trait of agronomic importance, as well as the plasticity of varieties under a variable environment influence [1,20].

The development of water-stress-tolerant cultivars of the common bean is a promising strategy to increase food security in marginal areas [21–23]. Drought-tolerance breeding programs usually select the best genotypes based on grain yield produced under water-deficit conditions [20]. However, evaluating the physiological and morphoagronomic traits in a combined approach may contribute to deepen the understanding of drought tolerance mechanisms [1,11]. In this sense, the objective of this study was to verify the influence of water deficit on physiological and morphoagronomic traits of common bean genotypes with contrasting drought tolerance.

## 2. Material and Methods

### 2.1. Plant Material and Experimental Design

The experiment was conducted in a greenhouse of the Agronomic Institute of Paraná (IAPAR) in Londrina, Paraná, Brazil (23°23′ S, 51°11′ W and 585 m altitude). A randomized complete block design in a $4 \times 2$ factorial scheme was used, consisting of four common bean cultivars under two water regimes (with and without water deficit), with six replications. The bean cultivars were of the Mesoamerican gene pool, two characterized as drought-tolerant (IAPAR 81 and BAT 477) and two as drought-sensitive (IAC Tybatã and BRS Pontal) (Table 1).

The seeds were pre-germinated in plastic incubator boxes containing filter paper moistened with distilled water. After germination, five seeds were transferred to pots containing 9 kg of soil (Red Latosol) and sand substrate at a 2:1 ($v/v$) ratio, to which 43.5 mg of the fertilizer mixture 04–30–10 ($N–P_2O_5–K_2O$) was added. The plants were thinned at development stage V3 (first fully expanded trifoliate leaves) to only one plant per pot.

**Table 1.** Characteristics of common bean cultivars evaluated under water-stressed and unstressed conditions.

| Cultivar | Origin | Market Group | Cycle (Days) | Growth Habit | Drought Reaction | Reference |
|----------|--------|--------------|--------------|--------------|------------------|-----------|
| BAT 477 | CIAT | Brown | 94 | III | Tolerant | [1] |
| IAPAR 81 | IAPAR | 'Carioca' | 92 | III | Tolerant | [24] |
| IAC Tybatã | IAC | 'Carioca' | 95 | II | Sensitive | [25] |
| BRS Pontal | EMBRAPA | 'Carioca' | 95 | II | Sensitive | [26] |

IAC: Agronomic Institute of Campinas; EMBRAPA: Brazilian Agricultural Research Corporation; IAPAR: Agronomic Institute of Paraná; and CIAT: International Center for Tropical Agriculture.

### 2.2. Induction and Monitoring of Water Deficit

Plants were cultivated at a water-holding capacity of 80% up to the phenological stage R5 (appearance of first flower buds), when water deficit in the respective plots was initiated, with a water regime of 30% of the water-holding capacity for 12 days. The substrate moisture was monitored with a real-time gravimetric system with automatic water replacement, according to the water regime established for each treatment. A drip irrigation system was installed, with self-compensating drippers (flow rate 2 L h$^{-1}$) coupled to the irrigation hoses. Maximum, minimum, and mean temperature variations within the greenhouse and relative humidity were measured by a thermohygrometer.

### 2.3. Morphoagronomic Evaluations

After collecting the plants at R9 (physiological maturation), the root volume (RV, in cm$^3$) was determined by the displaced water volume in a graduated beaker, when the fresh roots were inserted. Then, the shoots and root system were separately wrapped in paper bags and dried in a forced-ventilation oven, at 60 °C, for 72 h, to determine the shoot dry biomass (SDB, in g) and root dry biomass (RDB, in g). The number of pods per plant (PP), total number of grains per plant (TNG), number of grains per pod (NGP), 100-grain weight (GW, in g), and grain yield per plant (GY, in g) were also measured.

### 2.4. Physiological Evaluations

The net photosynthesis rate ($A$, in μmol $CO_2$ m$^{-2}$ s$^{-1}$), transpiration ($E$, in mmol $H_2O$ m$^{-2}$ s$^{-1}$), stomatal conductance ($g_s$, in mol m$^{-2}$ s$^{-1}$), intracellular concentration of $CO_2$ ($iC$, in μmol $CO_2$ mol$^{-1}$), and leaf temperature (Lt, in °C) were measured with an infrared gas analyzer (IRGA) (LCpro–SD, ADC BioScientific, Hoddesdon, UK), at a photon flux density of 1000 μmol m$^{-2}$ s$^{-1}$. The carboxylation efficiency (CE, in [(μmol m$^{-2}$ s$^{-1}$) (μmol mol$^{-1}$)$^{-1}$] was calculated by the $A$/$iC$ ratio and the intrinsic water-use efficiency ($iWUE$) (μmol $CO_2$ mol$^{-1}H_2O$) by the $A$/$g_s$ ratio.

The maximum photochemical efficiency of photosystem II ($F_v/F_m$) was determined by using a modulated fluorometer (MINI-PAM, Walz, Effeltrich, Germany), in which part of a leaf was placed in the dark for 30 min, held by specific clips. After the reading, the variable $F_v/F_m$ was determined, where $F_m$ is the maximum fluorescence intensity at which all photosystem II (FSII) reactions close; $F_0$ is the minimum fluorescence intensity when the FSII reaction centers are open; and $F_v$ is the variable fluorescence ($F_v = F_m - F_0$).

The relative leaf water content (RWC) was determined by the expression proposed by Weatherley [27]: RWC (%) = [(fresh biomass − dry biomass) × (turgid biomass − fresh biomass)] × 100. To determine the RWC, 2.0 cm$^2$ leaf discs from the main leaflet of the fifth trifoliate leaf were used to determine the fresh biomass on an analytical scale. Thereafter, the leaf discs were soaked in water for 24 h, to evaluate the turgid biomass. Dry biomass was determined after oven-drying of the material in a forced-air-circulation oven, at 60 °C, for 72 h.

All physiological traits were measured between 8:00 and 10:00 a.m., using the central leaflet of the third fully expanded trifoliol, for 0, 4, 8, and 12 days, with and without water deficit, for each replication.

### 2.5. Data Analyses

Analyses of variance (ANOVA) were performed after testing the homogeneity of variances and normality of the residuals by Bartlett [28] and Shapiro and Wilk [29] tests, respectively. Non-normal data or non-homogeneous variances were transformed by the Box and Cox [30] transformation. Two-way ANOVA was performed for the morphoagronomic traits, and three-way ANOVA was performed, with repeated measurements, for the physiological traits. The means were compared by the Scott–Knott [31] test at 5% probability. Pearson's linear correlation coefficient was analyzed by using the correlation network approach. Heatmap clustering was performed by Ward's [32] method based

on the standardized mean Euclidean distance. Statistical analyses were performed with R software (http://www.r-project.org) using 'ExpDes' [33], 'car' [34], 'qgraph' [35], and 'heatmaply' [36] packages.

## 3. Results

### 3.1. Morphoagronomic Traits

Variance analyses showed a significant effect ($p < 0.01$) of the genotype × water deficit interaction on the traits SDB, PP, TNG, and GY (Table 2). The sources of variation genotype and water-deficit variation were significant ($p < 0.01$) for all evaluated morphoagronomic traits. A comparison of the overall mean of all morphoagronomic traits under both water regimes showed that water deficit reduced all variables, except GW (Table 2).

**Table 2.** Analysis of variance, means, and coefficient of experimental variation (CV) of four common bean cultivars evaluated under water-stressed and unstressed conditions.

| Source of Variation | Mean Square of the Morphoagronomic Traits | | | | | | | |
|---|---|---|---|---|---|---|---|---|
| | **SDB** | **RDB** | **RV** | **PP** | **TNG** | **NGP** | **GW** | **GY** |
| Genotype (G) | 24.2 ** | 5.2 ** | 23.1 ** | 104.0 ** | 3181.7 ** | 10.2 ** | 77.5 ** | 234.4 ** |
| Water deficit (W) | 54.8 ** | 12.2 ** | 41.6 ** | 94.1 ** | 2882.1 ** | 14.1 ** | 137.5 ** | 137.1 ** |
| G × W | 33.2 ** | 2.1 ns | 5.8 ns | 40.0 ** | 1372.9 ** | 0.9 ns | 5.5 ns | 63.3 ** |
| Error | 5.6 | 1.9 | 5.8 | 4.4 | 245.1 | 0.6 | 3.5 | 9.7 |
| Mean (control) | 25.3 | 22.7 | 24.96 | 130.3 | 5.2 | 9.3 | 5.4 | 16.3 |
| Mean (water deficit) | 19.3 | 20.3 | 16.9 | 84.5 | 5.0 | 6.0 | 3.5 | 12.4 |
| CV (%) | 16.5 | 31.5 | 31.4 | 10.1 | 14.6 | 14.5 | 8.7 | 13.9 |

Shoot dry biomass (SDB); root dry biomass (RDB); root volume (RV); number of pods per plant (PP); number of grains per pod (SP); total number of grains per plant (TNG); 100-grain weight (GW); and grain yield per plant (GY); [ns] and ** not significant and significant at 1% probability by the *F*-test, respectively.

Under the water-deficit condition, the traits PP and NGP decreased significantly in all cultivars (Table 2 and Figure 1). On the other hand, the stress-sensitive and -tolerant cultivars did not differ statistically between the two water regimes for NGP. For cultivar IAPAR 81, considered tolerant to water deficit, no differences were observed between the water regimes for the traits GY, RV, and RDB, while for cultivar BAT 477, which is also tolerant, only GY and SDB did not differ statistically in both water regimes. On the other hand, the traits PP, TNG, RV, RDB, and SDB of the drought-sensitive cultivars (BRS Pontal and IAC Tybatã) were significantly affected by drought.

### 3.2. Physiological Traits

The analysis of variance indicated significant effects ($p < 0.05$) of the genotype × water deficit × time interaction on the traits $A$, $E$, $iC$, $i$WUE, and CE (Table 3). With regard to the other traits, double interactions were observed between genotype × water deficit ($F_v/F_m$), water deficit × time ($g_s$, RWC and Lt), and genotype × time ($g_s$). In the overall mean of both water regimes, water deficit reduced the traits $A$, $E$, $g_s$, $iC$, $F_v/F_m$, RWC, and CE. On the other hand, compared to the control, the variables $i$WUE and Lt increased under water deficit (Table 3).

The physiological evaluations over 12 days are shown in Figure 2. In general, the cultivars BRS Pontal and IAC Tybatã reduced the physiological activity considerably after the first days of water deficit, mainly for the traits RWC, CE, $A$, $iC$, $E$, and $g_s$. On the other hand, the variables of the cultivars IAPAR 81 and BAT 477 were higher at the end of 12 days of water deficit. Cultivar IAPAR 81 responded to water deficit with tolerance until the eighth day, and the physiological activities decreased after 12 days of evaluation, particularly for the traits $i$WUE, CE, $A$, $iC$, and $E$. Although many of the physiological traits of cultivar BAT 477 decreased on the fourth water-deficit day, the physiological activities were reestablished after the eighth day.

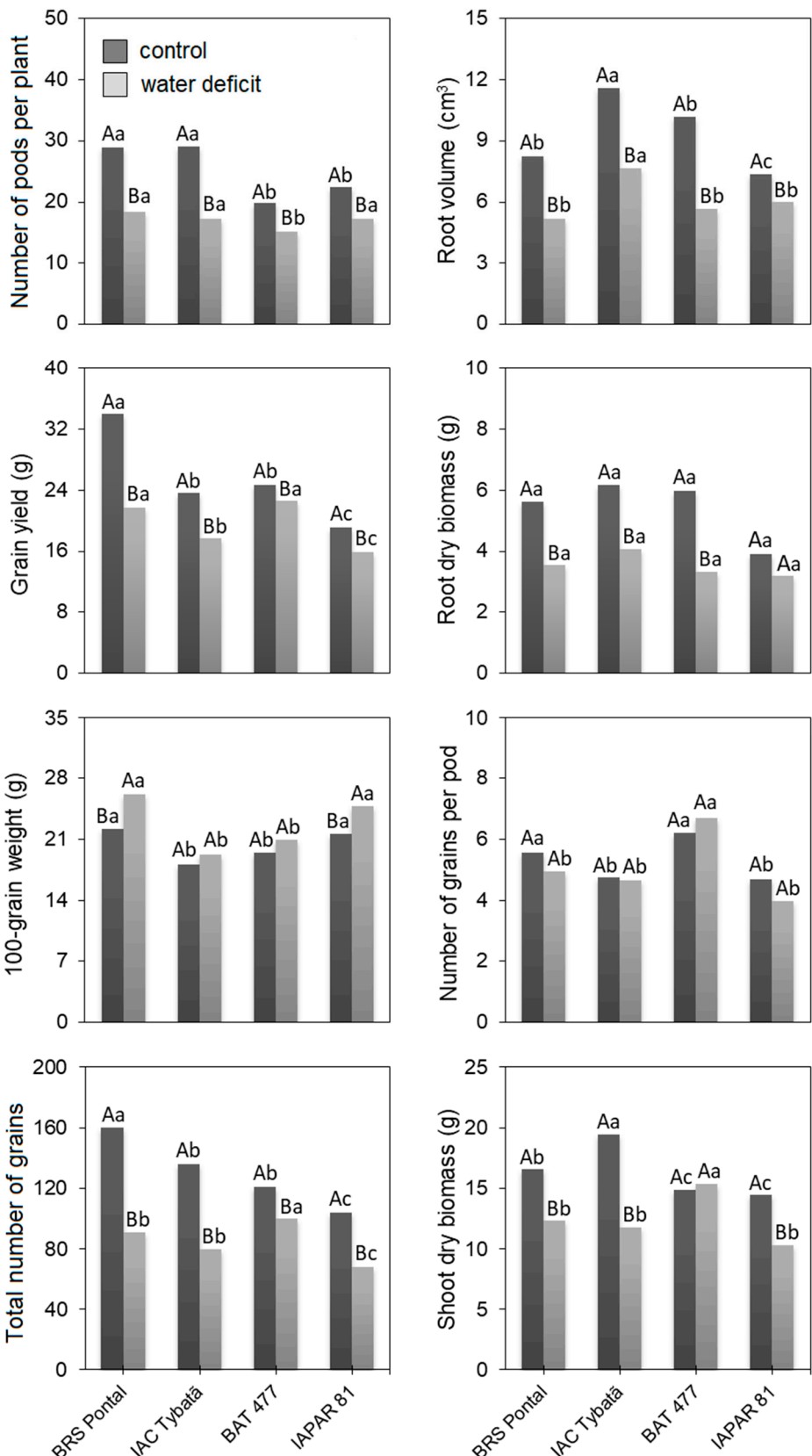

**Figure 1.** Means of morphoagronomic traits of common bean cultivars evaluated under control and water-deficit conditions. Uppercase and lowercase letters refer to comparisons by the Scott–Knott test ($p < 0.05$) between water regimes and cultivars, respectively.

**Table 3.** Analysis of variance, means, and estimates of experimental coefficients of variation (CV) of physiological traits of four common bean cultivars evaluated under control and water-deficit conditions during four periods (0, 4, 8, and 12 days).

| Source of Variation | Mean Square of Physiological Traits | | | | | | | | |
|---|---|---|---|---|---|---|---|---|---|
| | $A$ | $E$ | $g_s$ | $iC$ | $iWUE$ | $F_v/F_m$ | RWC | CE | Lt |
| Genotype (G) | 149.9 ** | 3.03 * | 0.28 ns | 0.65 ** | 133.6 ** | 2.39 ** | 768.2 ** | 558.0 ** | 1.69 ns |
| Water deficit (W) | 1867.4 ** | 81.88 ** | 4.98 ** | 5.54 ** | 32.2 * | 16.18 ** | 2902.8 ** | 123.9 ** | 102.51 ** |
| Time (T) | 691.2 ** | 28.90 ** | 6.63 ** | 2.04 ** | 213.7 * | 0.69 ns | 285.6 ** | 272.7 ** | 45.82 ** |
| G × W | 149.1 ** | 1.82 * | 0.07 ns | 0.55 ** | 151.8 ** | 2.38 * | 140.2 ns | 764.6 ** | 1.50 ns |
| W × T | 245.1 ** | 11.09 ** | 0.98 ** | 1.80 ** | 81.4 ** | 0.67 ns | 706.7 ** | 178.9 ** | 13.58 ** |
| G × T | 93.4 ** | 4.07 ** | 0.69 ** | 0.69 ** | 32.0 * | 0.33 ns | 107.2 ns | 126.6 ** | 2.10 ns |
| G × W × T | 40.3 ** | 1.93 * | 0.10 ns | 0.68 ** | 29.8 * | 0.47 ns | 117.7 ns | 91.1 ** | 0.92 ns |
| Error | 4.4 | 0.55 | 0.08 | 0.08 | 11.0 | 0.46 | 68.5 | 11.5 | 1.78 |
| Mean (control) | 16.4 | 3.6 | 0.29 | 220.1 | 70.2 | 0.80 | 80.9 | 223.7 | 29.0 |
| Mean (water deficit) | 10.1 | 2.3 | 0.19 | 109.9 | 73.3 | 0.78 | 73.1 | 115.8 | 30.4 |
| CV (%) | 15.7 | 25.1 | 36.3 | 27.3 | 32.1 | 5.8 | 10.7 | 4.8 | 5.5 |

Net photosynthesis rate ($A$, μmol $CO_2$ m$^{-2}$ s$^{-1}$); transpiration ($E$, mmol $H_2O$ m$^{-2}$ s$^{-1}$); stomatal conductance ($g_s$, m$^{-2}$ s$^{-1}$); intracellular concentration of $CO_2$ ($iC$, μmol $CO_2$ mol$^{-1}$); intrinsic water use efficiency ($iWUE$, μmol $CO_2$ mol$^{-1}H_2O$); maximum photochemical efficiency of photosystem II ($F_v/F_m$); relative leaf water content (RWC, %); carboxylation efficiency (CE, [(μmol m$^{-2}$ s$^{-1}$) (μmol mol$^{-1}$)$^{-1}$]); and leaf temperature (Lt, °C); ns, ** and *: nonsignificant and significant at 1% and 5% probability by the *F*-test, respectively.

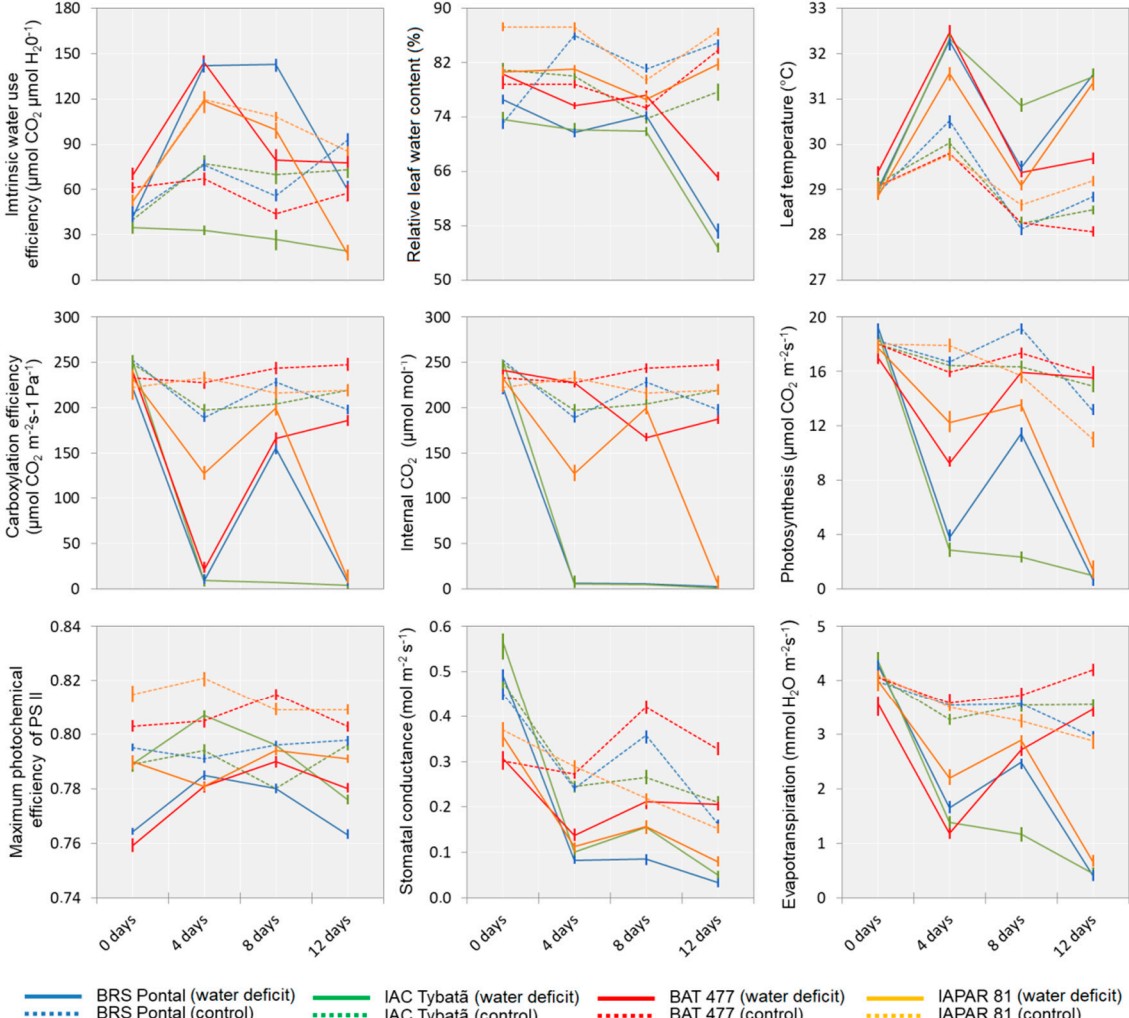

**Figure 2.** Physiological traits of common bean cultivars measured for 0, 4, 8, and 12 days, under control and water-deficit conditions.

### 3.3. Correlation Network

The correlation analyses between the morphoagronomic and physiological traits are shown in Figure 3. In comparison with the control condition (Figure 3a), positive correlations were observed between GY and TNG ($r$ = 0.91), SDB and $A$ ($r$ = 0.92), and SDB and $iC$ ($r$ = 0.95). Negative correlations of $i$WUE with the variables $E$ ($r$ = −0.97) and CE (r = −0.94) were also observed. Under the water-deficit condition (Figure 3b), the formation of a group of highly correlated ($r$ > 0.8) morphoagronomic (GY, TNG, PP, NGP, and RDB) and physiological traits ($i$WUE, $E$, $A$, $g_s$ and CE) was observed. The $i$WUE was positively correlated with GY ($r$ = 0.97), TNG ($r$ = 0.94), PP ($r$ = 0.88), RDB ($r$ = 0.87), and $E$ ($r$ = 0.87), and Lt negatively correlated with the variables $A$ ($r$ = −0.92), $E$ ($r$ = −0.90), $g_s$ ($r$ = −0.91), and $i$WUE ($r$ = −0.79).

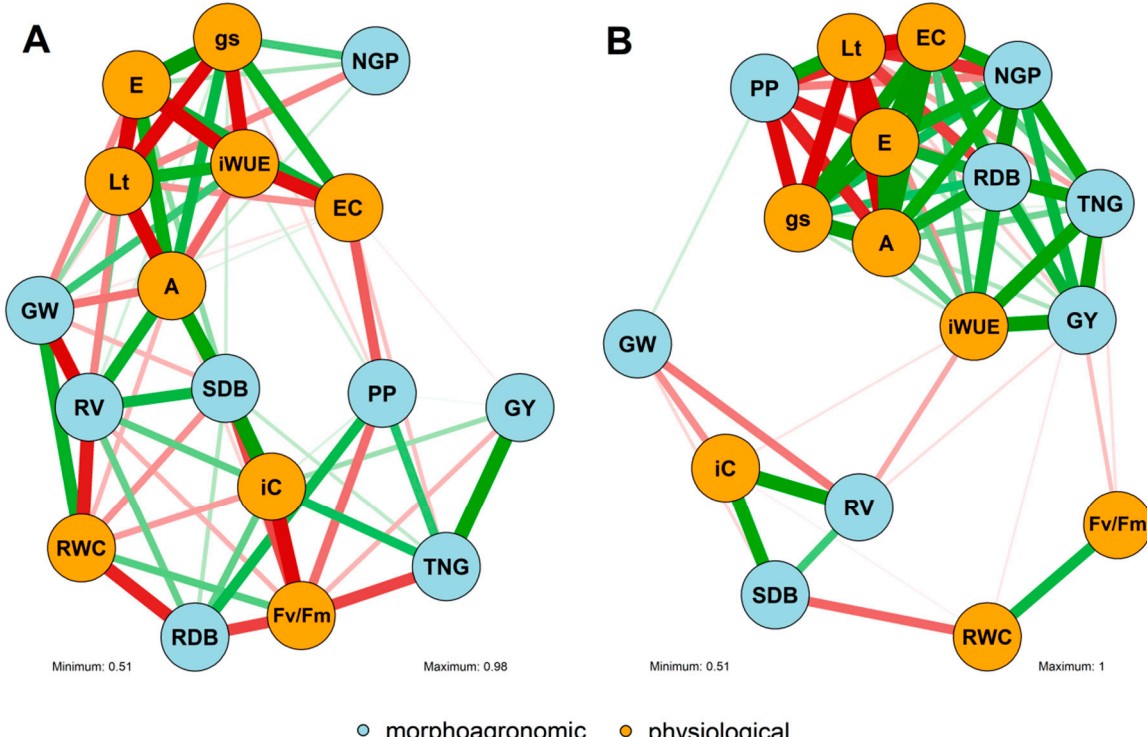

**Figure 3.** Network of correlations of morphoagronomic and physiological traits of common bean (*Phaseolus vulgaris* L.) cultivars grown under control (**A**) and water deficit (**B**). The green and red lines represent positive and negative correlations, respectively. The thickness of the lines is proportional to the strength of the correlation. Morphoagronomic traits: shoot dry biomass (SDB); root dry biomass (RDB); root volume (RV); number of pods per plant (PP); number of grains per pod (NGP); total number of grains per plant (TNG); 100-grain weight (GW); and grain yield per plant (GY). Physiological traits: net photosynthesis rate ($A$); transpiration ($E$); stomatal conductance ($g_s$); intracellular concentration of $CO_2$ ($iC$); intrinsic water-use efficiency ($i$WUE); maximum photochemical efficiency of photosystem II ($F_v/F_m$); relative leaf water content (RWC); carboxylation efficiency (CE); and leaf temperature (Lt).

### 3.4. Heatmap Representation

The heatmap clustering separated the treatments into two large groups (Figure 4). The water-stressed cultivars IAC Tybatã, BRS Pontal, and IAPAR 81 were allocated to the first group in which the values for morphological and physiological traits were generally the lowest evaluated, except for GW, Lt, and $iC$. The second group consisted of cultivars IAC Tybatã, BRS Pontal, and IAPAR 81, under control conditions, aside from cultivar BAT 477, under both water regimes.

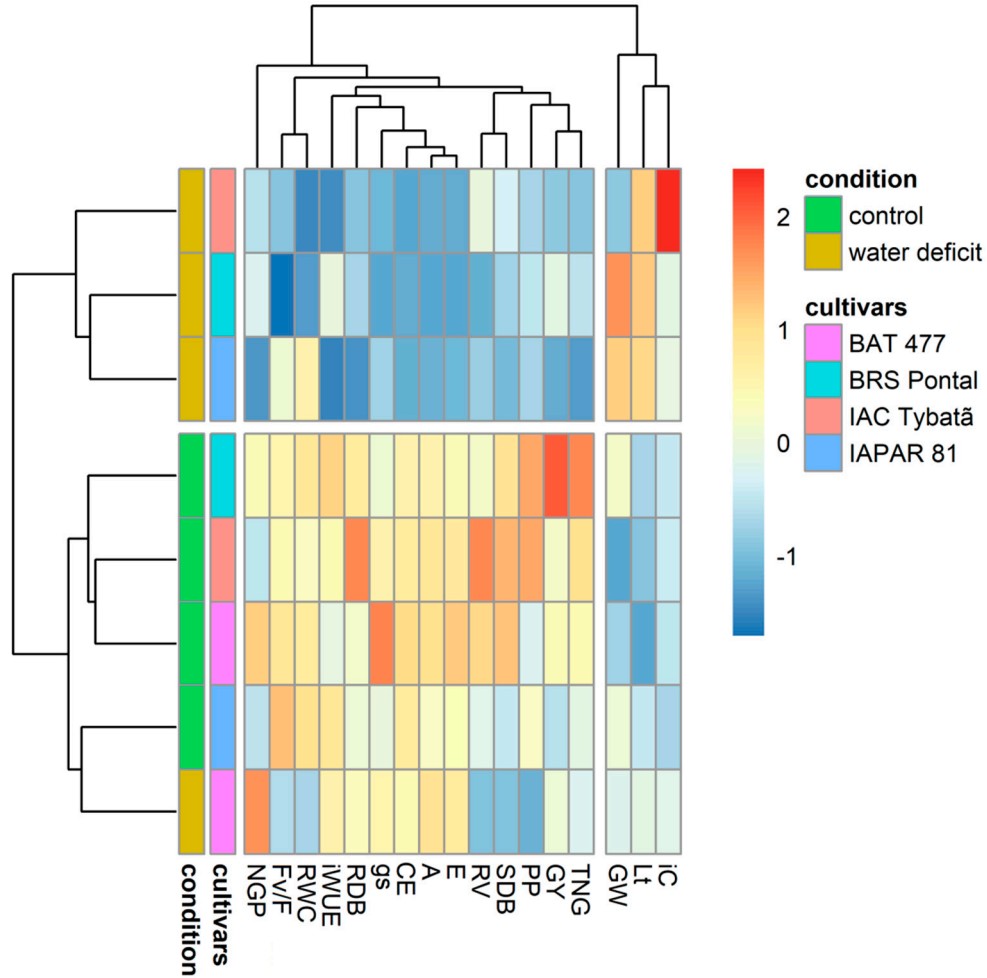

**Figure 4.** Heatmap of the morphoagronomic and physiological traits of common bean cultivars evaluated under water-stressed and unstressed conditions. Shoot dry biomass (SDB); Root dry biomass (RDB); root volume (RV); number of pods per plant (PP); number of grains per plant (NGP); total number of grains per plant (TNG); 100-grain weight (GW); grain yield per plant (GY); net photosynthesis rate (*A*); transpiration (*E*); stomatal conductance (*g$_s$*); intracellular concentration of $CO_2$ (*iC*); intrinsic water use efficiency (*i*WUE); maximum photochemical efficiency of photosystem II (*F$_v$/F$_m$*); relative leaf water content (RWC); carboxylation efficiency (CE); and leaf temperature (Lt).

## 4. Discussion

The significant effect of the genotype × water deficit interaction on morphoagronomic traits (SDB, PP, TNG, and GY) indicated a differentiated performance of the genotypes under control and water-deficit conditions. This finding is in agreement with results of Arruda et al. [1], Darkwa et al. [22], and Mazengo et al. [37], who also reported differential responses of common bean genotypes to stressful and non-stressful water regimes. In the literature, although water deficit is considered to be one of the main limiting factors for the common bean, several studies confirmed the wide genetic variability for drought tolerance [38–41].

In this study, all morphoagronomic traits, except GW, were negatively affected by water deficit. In an evaluation of the drought tolerance of 64 water-stressed common bean genotypes, Darkwa et al. [22] observed reductions of 2–29% in yield components. Similarly, Rao et al. [42] reported a mean GY reduction of 31% under drought compared to unstressed conditions, and Barrios et al. [43] confirmed that water deficit reduced the traits PP (63.3%), TNG (28.9%), and GW (22.3%). According to Assefa et al. [5], the yield components PP and TNG are the traits most negatively affected by water deficit.

The better adaptation of cultivar BAT 477 to water deficit has been attributed to a better root architecture system and, consequently, maximized water and nutrient uptake from the soil [1,44]. However, although cultivar BAT 477 stood out with the highest means for GY, NGP, TNG, and SDB under water deficit, the variables RV and RDB did not reach highest values. For cultivar IAPAR 81, although GY was not significantly reduced under water deficit, most of the evaluated morphoagronomic traits were affected. This cultivar had been released by IAPAR in 1997 and is still in use in Brazil, due to its broad adaptation and good tolerance to heat and water deficit [24,45].

For most physiological traits, the interaction between the factors genotype × water deficit × time was significant, indicating that, apart from the differential performance of the cultivars in relation to the water regimes, the water deficit duration also influences this performance. In an evaluation of the physiological response of common bean cultivars under water deficit, Rosales et al. [20] also reported the influence of water deficit period (13 and 22 days) on the physiological traits $A$, $E$, $g_s$, and $i$WUE. Similar results were observed by Rosales et al. [46] and Polania et al. [47]. According to Beebe et al. [1], physiological traits are important in the characterization of common bean genotypes for drought tolerance, for being intrinsically related to various mechanisms of water deficit adaptation.

Under water deficit, cascades of physiological responses are triggered to prevent water loss and plant death [1,20]. One of the first reactions of plants to water deficit is stomatal closure. In this way, the low stomatal conductance reduces water loss by transpiration, which decreases the $CO_2$ availability in the leaves and, consequently, the photosynthetic rate [48]. In addition, as a consequence of the decreased transpiration, the leaf temperature rises, which could decrease the photosynthetic rate to insufficient levels to replace the carbon used as substrate in the respiration process [49]. In this study, the $iC$, $E$, $g_s$, and $A$ rates of the drought-sensitive cultivars (IAC Tybatã and BRS Pontal) dropped drastically in the first days of water deficit, apart from the increase in Lt.

Cultivar BAT 477 behaved as drought-tolerant until the 12th and IAPAR 81 until the eighth day of water deficit. The drought-tolerance mechanisms of these cultivars were more efficient than those of the drought-sensitive cultivars, since their physiological activities were less affected by this stress. The tolerance of cultivars BAT 477 and IAPAR 81 is possibly caused by the absence of an abrupt drop in the amount of internal carbon in the first water-deficit days. Thus, the rubisco enzyme activity of these cultivars is more efficient than that of the sensitive cultivars. Consequently, the use of $CO_2$ is maximized, and the drop in the photosynthetic rate is mitigated [10,50]. In addition, RCW could reflect the plant-water status. IAPAR 81 kept this trait over time, but BAT 477 had a decrease in RCW after eight days of water deficit. Water statuses could cause different responses in each genotype, but it is not the only drought-adaption mechanism, as BAT 477 had a higher performance in morphoagronomic and physiological attributes.

The $i$WUE is indicated by the ratio of the photosynthetic rate by transpiration, and the measured values are related to the amount of carbon fixed by the plant per unit of lost water [51]. The $i$WUE of cultivar IAPAR 81 was not reduced until the eighth day of water deficit, while that of BAT 477 increased intensely until the 12th stress day. According to Rosales et al. [20], the $i$WUE of drought-tolerant genotypes is higher because they can overcome the limitation of $CO_2$ diffusion through the stomata by a more efficient mesophyll diffusion and effective $CO_2$ fixation. In this study, $i$WUE was positively correlated with yield components under water deficit. The importance of $i$WUE for the identification of water-deficit-tolerant genotypes has been pointed out by a number of authors [52–54].

The results of our study may contribute to a better understanding of the drought-tolerance mechanisms in common bean and guide breeders in the process of defining the main target traits in the selection of enhanced drought-tolerant genotypes. Cultivars considered drought-tolerant can be used as parents to develop improved populations with a high frequency of favorable genes involved in water-deficit tolerance. In addition, the identification of genotypes with contrasting drought tolerance is important in the choice of parents for crosses, to establish segregating populations for future mapping of quantitative trait loci (QTL) [4,55,56].

## 5. Conclusions

This study indicated that the drought tolerance of cultivar BAT 477 is not only a direct result of the low influence of stress on its yield components, but also a consequence of the participation of multiple adaptive physiological mechanisms, such as higher $i$WUE, $A$, $E$, CE, $g_s$, and $iC$ under water-deficit conditions. Cultivar IAPAR 81 can be considered drought-tolerant for short water-deficit periods only, since, after the eighth day of water deficit, the physiological activities decline drastically. The physiological characterization of common bean cultivars for water-deficit tolerance proved to be efficient in the identification of mechanisms underlying tolerance to this stress, and it can be used as an important tool in the selection of drought-tolerant bean genotypes.

**Author Contributions:** Conceptualization, V.M.-C., D.S.A., and J.P.T.; methodology, V.M.-C., D.S.A., J.P.T., and L.G.A.; data curation, D.M.Z.; formal analysis, D.M.Z.; funding acquisition, V.M.-C.; investigation, L.G.A., D.S.A., and J.P.T.; methodology, L.G.A., V.M.-C., D.S.A., and J.P.T.; project administration, V.M.-C.; resources, V.M.-C. and J.P.T.; software, D.M.Z.; supervision, V.M.-C. and J.P.T.; validation, V.M.-C. and J.P.T; writing—original draft preparation, D.M.Z.; writing—review and editing, D.M.Z, V.M.-C., and J.P.T. All authors have read and agreed to the published version of the manuscript.

**Funding:** This research was supported by Instituto Agronômico do Paraná (IAPAR) and Coordenação de Aperfeiçoamento de Pessoal de Nível Superior (CAPES).

**Conflicts of Interest:** The authors declare no conflicts of interest.

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
