# Peer review of "Effect of Water Deficit on Morphoagronomic and Physiological Traits of Common Bean Genotypes with Contrasting Drought Tolerance"

_water, doi:10.3390/w12010217_

Round 1

Reviewer 1 Report

Dear authors,

I have carefully revised your mansucript, and, on overall, it is really well written, explained and discussed. It is not easy to find a mansucript as optimum as it is, and, as far as I am concerned I will suggest its acceptance in the present form.

Very minor comments and suggestions have been included in the attached file. Please, consider them in order to improve the final result.

Congratulations for your work!

Author Response

Reviewer 1:

[L20-21]: You are going to develop an experience in order to evaluate the physiological and yield response of four cultivars to water stress. However, before of development, you know what of them are drought-tolerant... So, What is the contribution of this experience???

Answer: BAT 477 is well known as a tolerant cultivar (Sponchiado et al., 1989; White and Castillo, 1992; Terán and Singh, 2002), while IAPAR 81, IAC Tybatã and BRS Pontal were characterized by observation of reduction index (production) in Brazilian breeding programs. The comparation of physiological and yield traits in time course has not been performed before.         

Additional review was performed in text, as suggested.

Thank you for the contribution.

References

Sponchiado BN, White JW, Castillo JA, Jones PG. 1989. Root growth of four common bean cultivars in relation to drought tolerance in environments with contrasting soil types. Experimental Agriculture 25: 249–257.

White JW, Castillo JA. 1992. Evaluation of diverse shoot genotypes on selected root genotypes of common bean under soil water deficits. Crop Science 32: 762–765.

Terán H, Singh SP. 2002. Comparison of sources and lines selected for drought resistance in common bean. Crop Science 42: 64–70.

Reviewer 2 Report

Interesting research paper attempting to describe the yield components, morphoagronomic and physiologic traits of 4 bean cultivars submitted to a period of "stress". Nice experimental design.

But there are some weaknesses : 

It should be first pointed that the physical constraint must be qualified as "water deficit", whereas the plant stress is rather a consequence of the constraint applied. Moreover, it is likely that what is termed "water stress" occurs at different times, being for example more rapidly reached where stomatal conductance is insufficiently reduced, and therefore the water status more quickly affected. 

Secondly, the abstract is a bit insufficiently developed, in terms of morpho-physiological traits.

Thirdly, the temporal variations of the physiological traits suffer from irregularities (eg between days 0 and 4), which will need some comments in the discussion.

Finally, some figures (or captions) will need some corrections, before final manuscript acceptance.

Beyond these remarks, the paper is well written, concise, and well illustrated.  

In detail : 

L23 : traits have been measured

L39 and following : see above, what is said about water stress ; Line 39 should begin with : The effect of water deficit on common bean … 

L52 (en end of introduction): the water-use efficiency is also a trait of agronomic importance, as well as the plasticity of varieties facing a variable environment. 

L81 / At what date, probably depending on the cultivar, was the 30% water holding capacity reached?  

L87 : all the morphoagronomic traits have been measured (not evaluated)

L 96 : transpiration, not evapotranspiration!! 

L96-101 : Please specify how these physiological measurements were made : which Leaflet, which leaf rank, hour of measurement, nr of replicates, etc.

L102 : Instead of Phv/Phm, Fv/Fm is more frequently used in literature.

L142 - 148 : should be related to Table 2 and Figure 1 (where differences between bean cultivars are illustrated). It could be interesting to comment a little bit more the reason for G x S significant interactions (SDB, PP, TNG, GY)

L 150 : abbreviations used for traits could be added to the sub-plots' y axes

Table 3 : many significative G x T interactions reflect a possible discrepancy in the time necessary to induce water stress, and its secondary conséquences. From this point of view, it is a pity that the water potential has not been measured (or real water holding capacity along the trial).

Bottom of the Table (means) and caption : units must be reminded. 

L165 : What is liquid photosynthesis?? (net photosynthetic rate ?) ; Correct evapotranspiration --> leaf transpiration

L 173 : highest --> higher

Figure 2 and Line 179 : Plant physiological traits have been measured (not evaluated, even if a sampling has occurred). Some sharp variations seem to occur along time, without any comment of the authors: eg. Lt, CE iC (on day 4). Title of the Figure should be "Physiological traits … Under control and water deficit conditions".

L183 & 186 : correct Figure 1a and 1b --> 3A and 3B respectively

L 184 : Correct TGN -->  TNG

L193-184 : inversion between red and green : red is for negative corrélations, green for positive ones. In the correlations graphs EC should be written "CE". 

L202 and the following. The main utility of the heatmap is mainly to distinguish the BAT 477 behavior under water deficit conditions. 

L 213 : same corrections as those mentioned for Line 165

L 249 : the first "reaction" could be the plant water status itself (not measured); this reaction dépends on stomatal closure, which mitigates the water loss.  

L252 : Obscure meaning : the leaf temperature rise, which does not exceed 3°C, is doubtfully the main cause of the decrease in photosynthetic rates.

L 257-258 : it will be advisable to reason in terms of degradation over time  of the plant water statuses, with different morpho-agronomic and physiological consequences 

L266-268 : the increase in iWUE under water constraint is an interesting trait, but its beneficial agronomical value dépends on the capacity to form biomass! (the high ratio value can only be a consequence of a very low gs)

L280-287 : the perspective of breeding for drought tolerance should consider the question "which cultivar for which water deficit scenario". For example, IAPAR 81 could  interestingly chosen for cropping under irregularly rain-fed field conditions, and BAT 477, for more severe ones.

Author Response

Reviewer 2:

It should be first pointed that the physical constraint must be qualified as "water deficit", whereas the plant stress is rather a consequence of the constraint applied. Moreover, it is likely that what is termed "water stress" occurs at different times, being for example more rapidly reached where stomatal conductance is insufficiently reduced, and therefore the water status more quickly affected.

Answer: We agree with your point of view. Term modifications (e. g.  water stress to water deficit) were performed, as suggested.

[L81]: At what date, probably depending on the cultivar, was the 30% water holding capacity reached?

Answer: Unfortunately, this data was not collected in this study.

[L142-148]: It could be interesting to comment a little bit more the reason for G x S significant interactions (SDB, PP, TNG, GY)

Answer: It is reported in L227-234.

[L280-287]: The perspective of breeding for drought tolerance should consider the question "which cultivar for which water deficit scenario". For example, IAPAR 81 could  interestingly chosen for cropping under irregularly rain-fed field conditions, and BAT 477, for more severe ones.

Answer: We agree with your point of view, but these genotypes were used in this study for their academic value. Although they are not used as commercial cultivars, they could be used as parental genitors in common beans breeding programs.

Additional review was performed in text, as suggested.

Thank you for the contribution.